# Auramine O UV Photocatalytic Degradation on TiO$_2$ Nanoparticles in a Heterogeneous Aqueous Solution

Cristina Pei Ying Kong [1], Nurul Amanina A. Suhaimi [1], Nurulizzatul Ningsheh M. Shahri [1], Jun-Wei Lim [2,3], Muhammad Nur [4], Jonathan Hobley [5] and Anwar Usman [1,*]

1   Department of Chemistry, Faculty of Science, Universiti Brunei Darussalam, Jalan Tungku Link, Gadong BE1410, Brunei
2   HICoE-Centre for Biofuel and Biochemical Research, Institute of Self-Sustainable Building, Department of Fundamental and Applied Sciences, Universiti Teknologi PETRONAS, Seri Iskandar 32610, Perak Darul Ridzuan, Malaysia
3   Department of Biotechnology, Saveetha School of Engineering, Saveetha Institute of Medical and Technical Sciences, Chennai 602105, India
4   Center for Plasma Research, Integrated Laboratory, Universitas Diponegoro, Tembalang Campus, Semarang 50275, Indonesia
5   Department of Biomedical Engineering, National Cheng Kung University, No. 1 University Road, Tainan City 701, Taiwan
*   Correspondence: anwar.usman@ubd.edu.bn

**Abstract:** Amongst the environmental issues throughout the world, organic synthetic dyes continue to be one of the most important subjects in wastewater remediation. In this paper, the photocatalytic degradation of the dimethylmethane fluorescent dye, Auramine O (AO), was investigated in a heterogeneous aqueous solution with 100 nm anatase TiO$_2$ nanoparticles (NPs) under 365 nm light irradiation. The effect of irradiation time was systematically studied, and photolysis and adsorption of AO on TiO$_2$ NPs were also evaluated using the same experimental conditions. The kinetics of AO photocatalytic degradation were pseudo-first order, according to the Langmuir–Hinshelwood model, with a rate constant of $0.048 \pm 0.002$ min$^{-1}$. A maximum photocatalytic efficiency, as high as $96.2 \pm 0.9\%$, was achieved from a colloidal mixture of 20 mL (17.78 μmol L$^{-3}$) AO solution in the presence of 5 mg of TiO$_2$ NPs. The efficiency of AO photocatalysis decreased nonlinearly with the initial concentration and catalyst dosage. Based on the effect of temperature, the activation energy of AO photocatalytic degradation was estimated to be 4.63 kJ mol$^{-1}$. The effect of pH, additional scavengers, and H$_2$O$_2$ on the photocatalytic degradation of AO was assessed. No photocatalytic degradation products of AO were observed using UV–visible and Fourier transform infrared spectroscopy, confirming that the final products are volatile small molecules.

**Keywords:** Auramine O; basic dye; titania nanoparticle; photocatalytic degradation; mechanism

## 1. Introduction

Water is indispensable for sustaining the environment, keeping entire ecosystems regulated. Water is also an important natural resource and a vital asset for daily human life, as it is used for drinking, hygiene, and cooking, as well as in agriculture and fisheries. Although accessible clean water is crucially important over all regions, about one-sixth of the global population have difficulties in accessing clean water [1]. Additionally, it has also been claimed that four billion people are now facing a severe scarcity of clean water due to extinction, depletion, and pollution in major rivers of the world [2]. The presence of sediments, soil, and aquatic organisms, which are naturally produced by erosion of rock and soil, and the breakdown and rotting of organic matters, is related to water quality [3]. However, the presence of organic pollutants in water systems is even more dangerous, as they contaminate entire ecosystems and endanger human health. With this in mind, water

pollution continues to be one of the biggest issues humanity faces, especially with rapid population growth and increasing burdens from economic growth leading to increased industrial activity.

Although industrial pollutants are often handled in treatment or storage systems, they may eventually leach into the surrounding urban areas by rainfall, entering sewage systems and water reservoirs. This contamination degrades the quality of water, and has therefore been singled out as one of the reasons for economic slowdown in many developing countries [4]. One of the most serious classes of environmental pollutants found in water systems is synthetic dyes, which are used intensively in the dying process in the textile, paper, leather, plastic, and rubber industries, due to their vibrant colors and low cost [5,6]. Notably, over 50% of the global dye usage, and the resulting contamination, occurs in developing regions of Asia [7]. Even the presence of low concentrations of dyes can greatly affect aquatic life and ecosystems, in terms of eutrophication and perturbations. This is because of the color intensity of dyes, which has the ability to prevent penetration of sunlight through water, resulting in a clear decline in the rate of photosynthesis, lowering dissolved oxygen levels. This increases the biochemical oxygen demand [8]. Even though strict control of water quality has been implemented in many countries, in order to fulfil the regulation of minimum allowed concentrations of pollutants, better treatments to eliminate these persistent organic compounds from industrial wastewater must also be curated.

Typically, industrial dyes are highly water-soluble so that they are cheaper and easier to use in manufacturing dying processes [9–11]. These dyes are categorized as chromophoric or auxochromic dyes, containing different moieties and functional groups which are responsible for their color intensity [12]. Among them, Auramine Orange (AO) and its derivatives, which are cationic diarylmethane dyes having yellow fluorescence and vibrant color, are widely mass produced for use in food, textile, paint, ink, plastic, and cosmetic industries [13,14]. Due to AOs' carcinogenic nature, there have been studies that examined its biotransformation to reactive species in target organs of rats and humans when administered orally [15]. Considering that AO and its derivatives cause long-term impacts on aquatic environments, as well as causing other health risks [16,17], their removal from wastewater before discharge is imperative. The treatment of such effluents would not only protect the water systems and the entire ecosystem, but also encourage manufacturers to reuse the spent water from their dyeing processes.

A variety of methods, including electro-coagulation [18], chemical precipitation, coagulation and filtration [19], reverse osmosis membrane [20], ozonation [21], aerobic and anaerobic processes [22], adsorption on activated carbon [23], and photocatalytic degradation [24], have been devised for the treatment of industrial wastewater effluents. However, amongst these methods, adsorption and photocatalysis have attracted great interest due to their cost-efficiency, sustainability, and selectivity [25,26]. Heterogenous photocatalysis has been reported to be more desirable as this method shows several advantages in the decolorization of wastewater due to the high efficiency of photocatalytic degradation in the removal of dyes from complicated organic effluents [27,28], easy waste disposal, low cost, and complete mineralization [24]. Additionally, this process can be applied in ambient or mild pressure conditions, using solar energy for power, or pre-existing natural UV light in water purification systems, in order to degrade synthetic dyes completely into less harmful byproducts [29].

The key mode of action of heterogenous photocatalysis is the degradation of dyes during a chemical reaction with photochemically generated hydroxyl ($OH^{\bullet}$) and oxygen ($O_2^{-\bullet}$) radicals on the surface of the photocatalyst [30–32]. Therefore, the photocatalytic degradation of synthetic dyes is strongly governed by the dynamics of adsorption onto the catalyst surface and their reactivity with the radicals once there. This can be further controlled using several parameters, such as irradiation time, the catalyst dosage, dye concentration, and $H_2O_2$-mediated processes. Additionally, using nanoparticles (NPs) instead of micro-powders should greatly increase reaction rates and efficiencies.

The efficiency of dye photocatalytic degradation depends on their relative redox potentials with respect to those of the catalyst, allowing electron and hole transfer to generate $O_2^{-\bullet}$ and $OH^\bullet$ radicals [31]. In this sense, photocatalytic degradation of AO on TiO$_2$ NPs and its kinetics have been reported by Montazerozohori [33]. Photocatalysis of AO on semiconductor oxides has been intensively investigated [26,34], but several aspects of the photocatalytic degradation of the dye are still deficient. In particular, photochemical systems are complicated and it takes time to elucidate systems as the literature about them builds up. In general, a lot of works are required to generate a consensus as to what is actually going on.

Therefore, in this study, photocatalytic degradation of AO on anatase TiO$_2$ NPs under 365 nm light irradiation was investigated. The objective was to systematically evaluate the effect of irradiation time, the initial concentration of AO, and catalyst dosage on the photocatalytic degradation of the dye. The efficiency and rate constant of the photodegradation were estimated based on absorption spectra of AO before and after irradiation. The photocatalytic degradation data were analyzed with standard empirical models. The thermodynamics of the AO photodegradation process were assessed by monitoring the effect of temperature. The photocatalytic degradation mechanism was further assessed by observing the effect of pH and additional scavengers as well as H$_2$O$_2$.

This work should provide a baseline for future works, which may include using doped and sensitized TiO$_2$ in order to shift the absorbance further to the visible to improve catalytic efficiency [35–39]. It is important to highlight that there are several differences between this study and those in the literature, including the use of NPs, which should give better photocatalytic degradation rates. There are also several similarities and some agreements between this current study and the reported works, which give affirmative verification of many of the conclusions from the earlier work, which is a general duty of the traditional scientific approach.

## 2. Results

### 2.1. Photolysis and Adsorption of AO

The photolysis of 35.06 μmol L$^{-1}$ of AO in aqueous solution in the absence of TiO$_2$ NPs under 365 nm light irradiation is shown in Figure S1A. The absorption spectra of AO have two main peaks at 432 nm and 370 nm. The spectrum shows that the absorbance of AO solution gradually and slightly decreased over time when the AO solution was exposed to the UV irradiation, confirming that the dye was slowly decomposing. From these absorption spectra, the concentration of AO was extracted and plotted against the irradiation time, as shown in Figure S1B. It is clearly seen that the concentration of AO after 180 min of irradiation is only slightly lower than it was before irradiation, and hence, the efficiency of the noncatalytic photolysis was determined to be less than 8% even after such a prolonged irradiation time. This is conclusive evidence that the direct photolysis of AO by the 365 nm light irradiation is not efficient.

In comparison, adsorption of AO onto TiO$_2$ NPs was revealed by a gradual decrease in the absorbance of AO with contact time, as shown in Figure S2. Based on this decrease in absorbance up to 180 min of contact time, the adsorption efficiency of AO was found to be less than 3%, indicating that the adsorption of AO by TiO$_2$ NPs is also inefficient.

### 2.2. Photocatalytic Degradation of AO

Figure S3 shows a colloidal mixture of 20 mL AO solution with 2.5 mg TiO$_2$ NPs before and after UV light irradiation for 100 min. It can clearly be seen that the colloidal mixture was completely decolored, suggesting that 100% efficient photocatalytic degradation of AO had occurred within this time.

The degradation of the dye in the heterogeneous aqueous solution was then monitored at different irradiation times from 0 to 150 min, and the reduction in the color with increasing irradiation time is shown in Figure S4. The absorption spectra of the heterogeneous aqueous solutions of AO and TiO$_2$ NPs, after being exposed to the UV light, were measured

after centrifugation. These spectra are shown in Figure 1. The concentration of AO in the heterogeneous aqueous solutions was determined based on its absorbance at 432 nm, at which value the molar decadic extinction coefficient of the dye is 25,300 L mol$^{-1}$ cm$^{-1}$. The photodegradation efficiency ($\eta$), also known as color removal rate, was then calculated as

$$\eta(\%) = \frac{(C_0 - C_t)}{C_0} \times 100\% \tag{1}$$

where $C_0$ and $C_t$ are the initial and remaining concentration of AO in the mixtures at irradiation time, $t$.

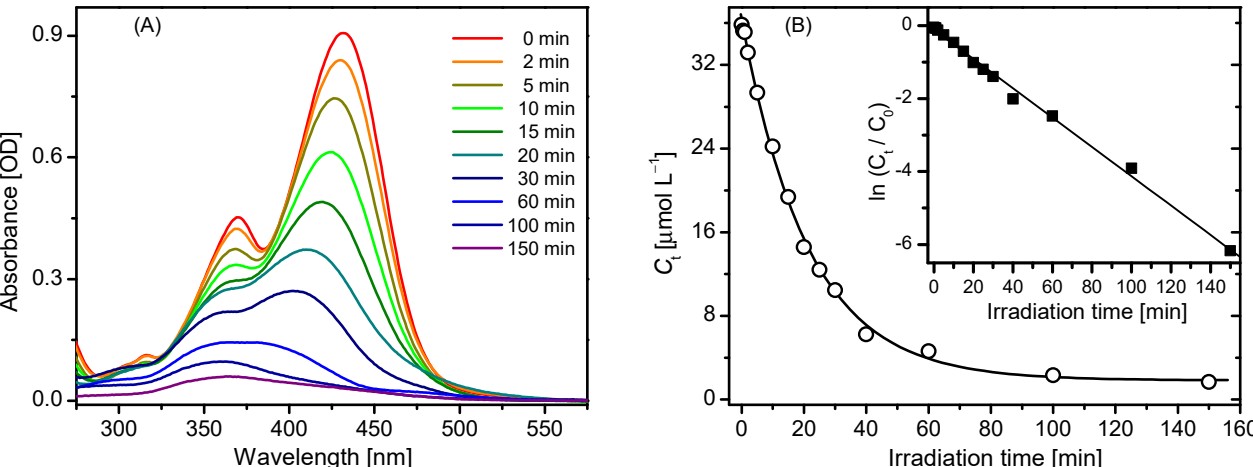

**Figure 1.** (**A**) Absorption spectra of AO (35.06 µmol L$^{-1}$) in an aqueous colloidal solution, in the presence of 5 mg TiO$_2$ NPs, after different irradiation times, as indicated; and (**B**) a plot of $C_t$ against AO as a function of irradiation time, simulated using the Langmuir–Hinshelwood kinetic model. Inset: the linear plot of $ln(C_t/C_0)$ against irradiation time.

It was found that the dye was almost completely degraded within 150 min of irradiation, with the $\eta$ value being 96.2 ± 0.9%. This is slightly higher than reported for methylene blue (MB) (93.1%) and rhodamine B (RhB) (96.1%) [28]. Considering that, alone, the non-photocatalytic photolysis and dark-adsorption of AO onto TiO$_2$ NPs were inefficient, the enhanced degradation of the dye in Figure 1 can be assigned to a photocatalytic process on the catalyst surface.

The photocatalytic degradation kinetics of AO were evaluated by simulating the experimental data using a single exponential decay function. Here, the degradation rate constant is considered to be linearly related to concentration of the dye, according to the Langmuir–Hinshelwood (L–H) model [40,41]. The L–H equation is expressed as [40–42]

$$C_t = C_0 \exp(-k_{obs}t) \tag{2}$$

where $k_{obs}$ is the observed degradation rate constant, and is determined from the single exponential decay of $C_t$ as a function of irradiation time, $t$.

As shown in Figure 1B, the data fit well with the L–H kinetic model, suggesting that the heterogeneous photocatalytic degradation of AO is a pseudo-first-order reaction. This is unambiguously supported by the linear correlation between $lnC_t/C_0$ as a function of irradiation time. From this best fit, the degradation rate constant, $k_{obs}$, of AO was estimated to be 0.048 ± 0.002 min$^{-1}$. In comparison, under the same experimental conditions, the $k_{obs}$ value of AO is much slower than those of RHB (0.115 ± 0.005 min$^{-1}$) and MB (0.173 ± 0.019 min$^{-1}$) [28].

The photocatalytic degradation of AO must be proportional to the external mass transfer of the dye onto the catalyst surface. In this sense, the mass transfer behavior of AO was analyzed using the intraparticle diffusion model, given by [43]

$$C_0 - C_t = k_i t^{1/2} + C \tag{3}$$

Here, $k_i$ is the diffusion rate and $C$ is the boundary layer thickness on the catalyst surface.

The simulation plot shown in Figure 2A demonstrates that mass transfer occurred in three diffusion steps. There was slow diffusion with a $k_i$ of 0.89 µmol L$^{-1}$ min$^{-1/2}$ which occurred within 1 min of irradiation, and is associated with early diffusion of AO onto the catalyst surface. This was followed by a fast and effective diffusion with a $k_i$ of 5.69 mmol L$^{-1}$ min$^{-1/2}$. Finally, another slow diffusion step occurs, with a $k_i$ of 0.89 µmol L$^{-1}$ min$^{-1/2}$ at irradiation times longer than 40 min until the complete degradation of AO in the solution is achieved. It is noteworthy that extrapolation of the simulation plot at early irradiation times passed the origin, implying that boundary layer thickness can be assumed to be negligible. In other words, the diffusion rate of the dye in the solution was comparable to that on the catalyst surface. This also provides an interpretation that photodegradation byproducts of AO did not disturb the diffusion of the dye onto the catalyst surface.

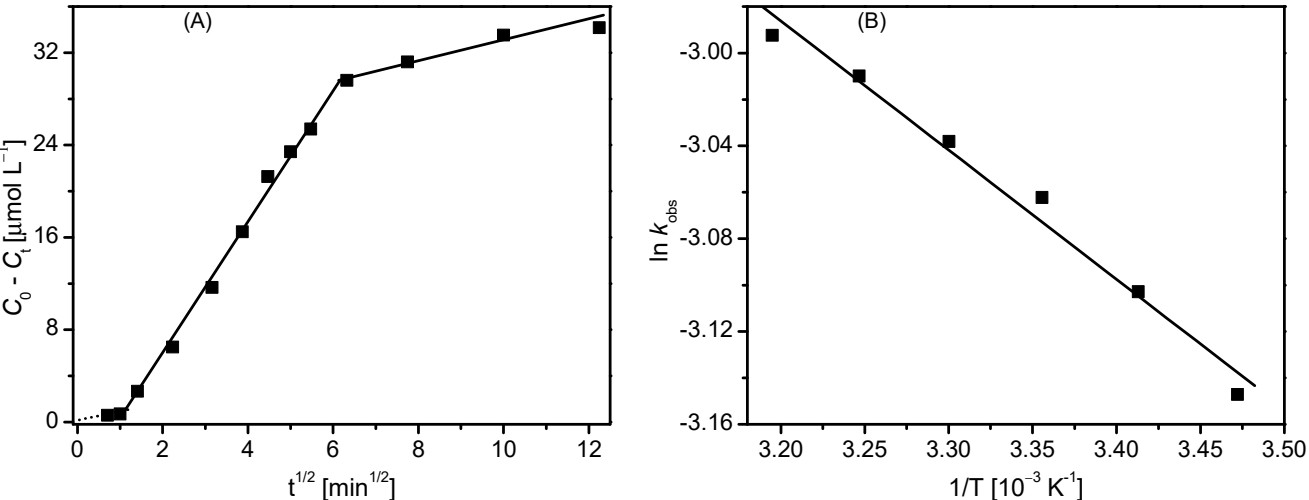

**Figure 2.** (**A**) Simulation of kinetic data with the intraparticle diffusion model, and (**B**) an Arrhenius plot of $lnk_{obs}$ against 1/T for the photocatalytic degradation of AO in the heterogenous aqueous solution. The lines are the best fits. The activation energy of the photocatalytic degradation was deduced based on the Arrhenius equation.

*2.3. Effect of Temperature*

The photocatalytic degradation of AO depends on diffusion and immobilization of the dye onto the TiO$_2$ NPs; hence, it should be affected by temperature. Figure S5 shows absorption spectra of AO solutions before and after UV light irradiation for 30 min at different temperatures. The effect of temperature was then further analyzed based on the $k_{obs}$ of AO photocatalytic decomposition. These results demonstrated that $k_{obs}$ increased with the temperature, suggesting that diffusion and immobilization of the dye on the catalyst surface were accelerated at higher temperature. Additionally, electron–hole recombination is also believed to accelerate with increased temperature [27,44–46].

The activation energy ($E_a$) of the photocatalytic degradation of the dye was then evaluated based on the effect of temperature (15–40 °C) on $k_{obs}$ by using the Arrhenius equation;

$$k_{obs} = A \exp(-E_a/RT) \tag{4}$$

where $A$ is the pre-exponential factor, $R$ is the gas constant, and $T$ is the temperature.

Based on the Arrhenius plot of $lnk_{obs}$ as a function of $1/T$ shown in Figure 2B, the $E_a$ of photocatalytic degradation of AO on the TiO$_2$ NPs was estimated to be 4.63 kJ mol$^{-1}$. For comparison, under the same experimental conditions, the $E_a$ value of the photocatalytic degradation of MB was 37.3 kJ mol$^{-1}$ [27]. Thus, the potential barrier of the photocatalytic degradation of AO on the catalyst surface is much lower than that of MB. Therefore, it can be concluded that the oxidation reaction between AO and the generated O$_2^{-\bullet}$ and OH$^{\bullet}$ radicals on the catalyst surface is much more energetically favorable than it is for MB.

## 2.4. Effect of Various Parameters on the Photocatalytic Degradation of AO

As this photocatalytic degradation is an oxidation reaction of the dye on the catalyst surface, at a certain temperature, the reaction should depend on various parameters, including the dye concentration and catalyst dosage. Figure S6 shows the spectra of AO with different initial AO concentrations ($C_0$) before and after photocatalytic degradation. Based on these spectra, the concentration of AO that was degraded during the photocatalysis, and the $\eta$ value, increased and reached an optimum condition with respect to concentration ($C_0$). This was followed by a nonlinear decrease, as seen in Figure 3. This finding highlighted that the photocatalytic activity of the dye is related to the number of dye molecules in the heterogeneous colloidal mixture, and the low $\eta$ value at high initial concentration is attributed to the well-known screening effect [47–49].

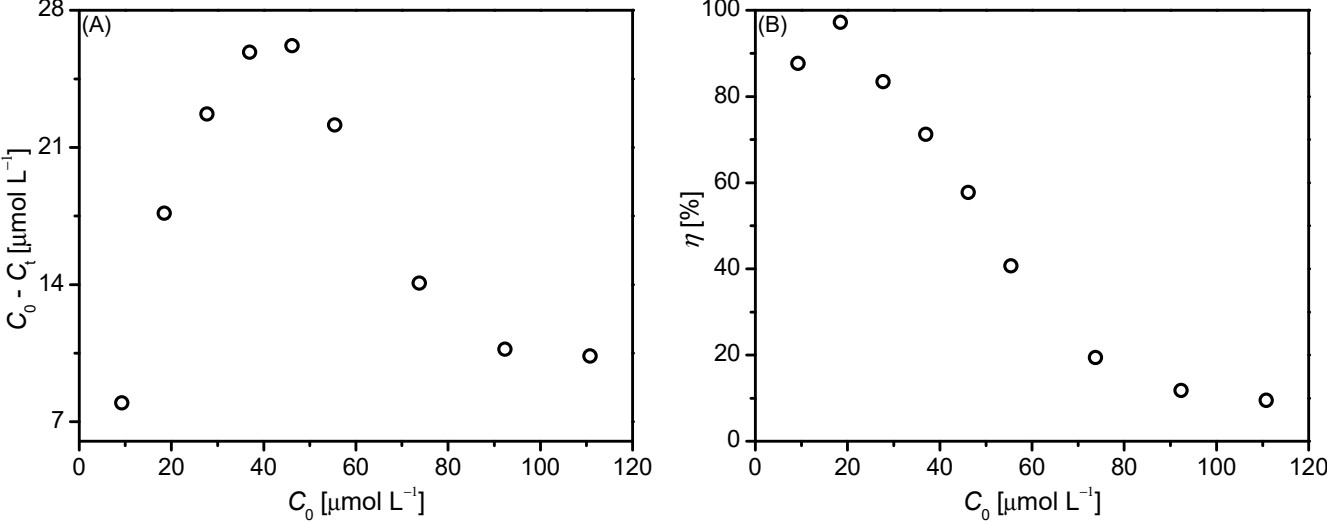

**Figure 3.** (**A**) Plot of $C_0 - C_t$; and (**B**) the $\eta$ values of photocatalytic degradation of AO as a function of the initial AO concentration ($C_0$) in the colloidal mixture with 5 mg TiO$_2$ NPs after irradiation for 30 min.

As shown in Figure S7, the catalyst dosage also affects the photocatalytic degradation of AO. Based on the absorption spectra of AO before and after irradiation in the presence of different dosages of TiO$_2$ NPs, the degradation efficiency was found to decrease nonlinearly with the catalyst dosage. This phenomenon is assigned to the inefficient photocatalytic degradation of the dye at high catalyst dosages.

The photocatalytic degradation of AO was also followed in the presence of a small amount (1–5%) of benzoquinone (BQ) and tert-butanol (t-BuOH) which scavenge O$_2^{-\bullet}$ and OH$^{\bullet}$ radicals, respectively. It was found that the $\eta$ value of AO decreases abruptly with the addition of BQ and t-BuOH, as shown in Figure 4A. This result confirms that the degradation mechanism by UV/TiO$_2$ NPs depends on the oxidation reaction of the dye with both O$_2^{-\bullet}$ and OH$^{\bullet}$ radicals, as has been described in several studies [50,51]. The formation of O$_2^{-\bullet}$, by reduction of solvated oxygen in the aqueous solution, is an important step to prevent the recombination of the photogenerated electrons and holes [52]. High concentrations of oxygen in the solution should reduce the recombination process and hence assist the formation of both O$_2^{-\bullet}$ and OH$^{\bullet}$ radicals. To explore this possibility, the

effect of adding a small amount of $H_2O_2$ on the $k_{obs}$ of the photocatalytic degradation of AO was evaluated, as presented in Figure 4B. The dissociation of $H_2O_2$ enhances the concentration of oxidants. This accelerates the generation of $O_2^{-\bullet}$ and $OH^{\bullet}$ radicals [53], leading to a higher photodegradation rate, although the efficiency of AO after prolonged irradiation time was almost unchanged (96.2–97.3%). Thus, the results further suggest that the generation of $O_2^{-\bullet}$ and $OH^{\bullet}$ radicals is the rate-determining step of the photocatalytic degradation of the dye.

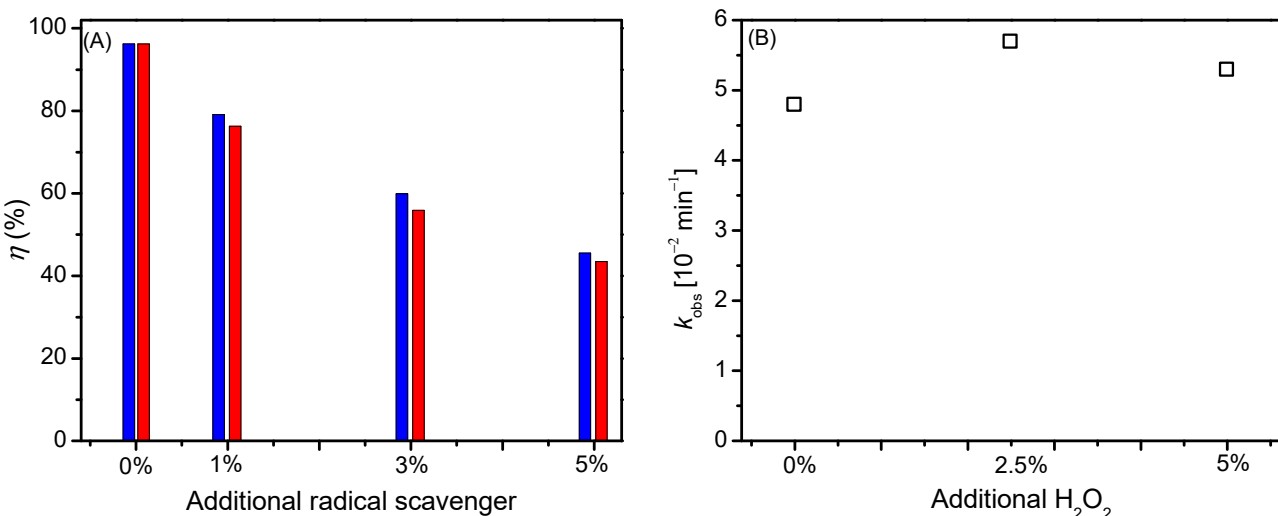

**Figure 4.** Plots of $\eta$ values of photodegradation of AO (35.06 µmol $L^{-1}$) in aqueous colloidal solution in the presence of 5 mg $TiO_2$ NPs after irradiation for 30 min with the addition of (**A**) BQ (blue bars) and t-BuOH (red bars), and (**B**) $H_2O_2$.

The effect of the pH of the medium on the $\eta$ value of photocatalytic degradation of AO is shown in Figure S8. At pH lower than 9, the $\eta$ value increased with pH. The $\eta$ value reached a maximum value at pH 8–9, and then abruptly decreased at pHs above 10.

### 2.5. FTIR Analysis

Steady-state FTIR spectroscopy was used to search for large molecular fragmentation of the products from the photocatalytic degradation of AO. For this analysis, the AO solution after irradiation (see Figure S3) was collected and dried. The vibrational spectrum was then measured in the spectral range of 4000 to 450 $cm^{-1}$, as shown in Figure 5. For comparison, the spectrum of AO before irradiation is also presented. The main vibrational bands of AO before irradiation were observed at 3407, 3004, 1691, 1602, 1374, 1156, 941, and 821 $cm^{-1}$, which are assigned to NH stretching of dimethyl amine, C=N stretching, CH of aromatic rings, C=C stretching of aromatic rings, CH bending of aromatic rings, C–N stretching, C–C stretching, and CH out-of-plane bending vibrations of the dye, respectively. Similar spectral features of AO were reported by Mallakpour et al. [54].

It is important to note that the FTIR spectrum of AO after irradiation is similar to that before irradiation. No new additional bands are clearly observed, except a broad band at 600–900 $cm^{-1}$ which could be assigned to the symmetric stretching vibrations of O–Ti–O of anatase $TiO_2$ NPs [55] remaining after the photocatalysis. This confirms that the steady-state FTIR spectroscopy did not detect any photoproducts of AO; instead, it detected the remaining AO and $TiO_2$ NPs. This provides an interpretation that either the photocatalytic products have low infrared cross sections or they are volatile and evaporated during the photoirradiation or drying process, so that none of them were detected by the steady-state measurement. In support of this latter argument, UV–visible spectra of AO (and many other reported dyes) only show a reduction in absorption of AO, with no new peaks being identifiable as coming from any large fragment degradation products.

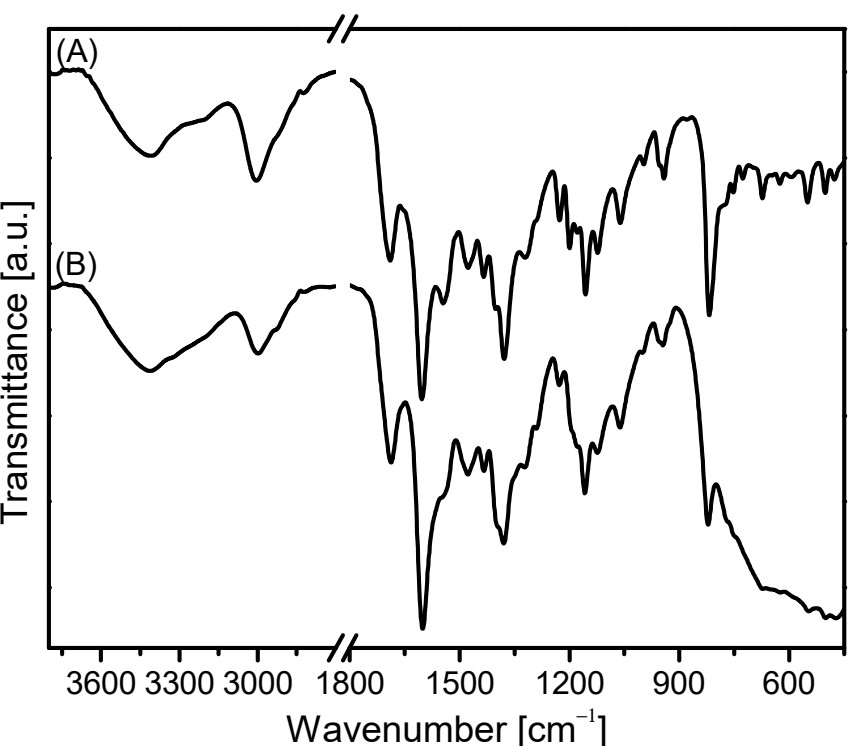

**Figure 5.** FTIR spectra of AO (**A**) before and (**B**) after photocatalytic degradation.

## 3. Discussion

In this study, the TiO$_2$ NPs used as photocatalyst were pure anatase crystals, which was confirmed based on FTIR spectra (not shown) and XRD patterns (see Figure S9). The particle size of TiO$_2$ NPs is approximately 100 nm with a BET surface area and pore volume of 12.791 m$^2$ g$^{-1}$ and 0.05733 cm$^3$ g$^{-1}$, respectively [27]. With the bandgap energy being 3.20 eV, the 365 nm light irradiation easily excites the TiO$_2$ NPs, generating electron–hole pairs [56]. This is an advantage in the degradation of AO, because AO absorbs mainly in the visible region. In this sense, the UV irradiation mostly excites the catalyst and, in any case, the photolysis of the dye is inefficient.

As has been discussed in many studies, separation and migration of the photo-generated charge carriers onto the catalyst surface is essential for the photocatalysts to generate the OH$^\bullet$ and O$_2^-$$^\bullet$ radicals. With this in mind, the anatase phase of TiO$_2$ has been theoretically and experimentally revealed to possess high charge-carrier mobility and low charge resistance [55,57], and hence it has high potential as photocatalyst. The photocatalytic degradation of organic dyes is not only governed by the formation rate of OH$^\bullet$ and O$_2^-$$^\bullet$ radicals on the catalyst surface. It should also be governed by the diffusion and immobilization of dyes onto the catalyst surface as well as by the potential energy barrier of the oxidation reaction of the dyes.

The diffusion of a dye in the colloidal solution depends on its hydrodynamic size (related to its molecular structure) as given by the Einstein–Stokes relation. Although there is no report of this for AO in the literature, the structure and size of MB and RhB are approximately comparable to AO. Therefore, the diffusion constant ($D$) value of AO in aqueous solution can be expected to be close to those of MB ($6.74 \times 10^{-6}$ cm$^2$ s$^{-2}$) [58] or RhB ($4.50 \times 10^{-6}$ cm$^2$ s$^{-2}$) [59]. The $D$ value is positively related to the diffusion-limited rate constant ($k_D$) by the generalized Smoluchowski equation, as given by

$$k_D = 4\pi\sigma D \tag{5}$$

where $\sigma$ is the encounter distance. As $k_{obs}$ is a proportionally related to $k_D$, the $k_{obs}$ value of AO can be considered to be comparable to those of MB and RhB. In fact, it was at least two

or three times lower in this case compared to those of MB and RhB. A plausible reason for the lower than expected $k_{obs}$ value measured for AO is its nonplanar structure, which is due to the few degrees of rotation of both *N*,*N*-dimethylaniline moieties with respect to their C−C bond. In order to examine the planarity of AO, ab initio structural optimization was performed using Gaussian basis sets in Chem3D. The structure was optimized for 500 iterations, and the minimum RMS gradient was 0.1. As shown in Figure 6, the MM2 force field suggests that the structure of AO in the gas phase is nonplanar with a torsion angle between the two aromatic ring systems being ~40°. A similar result was also observed using an MMF94 force field, but the torsion angle of the optimized structure was larger (~65°). Although the polar environment in aqueous solution might suppress the rotation of aromatic rings and alter the charge redistribution on AO, the calculation suggests a relatively nonpolar structure of AO. The torsional dynamics could cause intramolecular charge redistribution on AO, leading to strong to medium intermolecular friction, thus slowing the dynamics of the photocatalytic reaction.

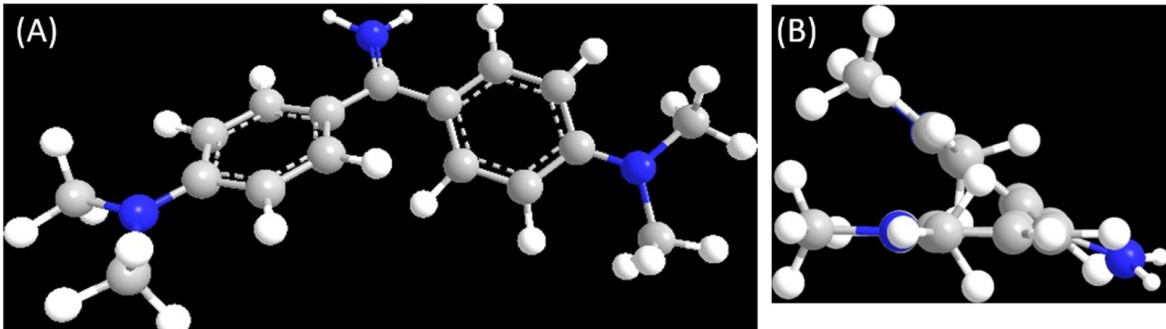

**Figure 6.** (**A**) Nonplanar structure of AO optimized using Chem3D (the force field MM2), and (**B**) the torsion angle of 40° between the two aromatic systems.

The photocatalytic degradation behavior of AO should also be considered based on the driving force to immobilize the dye on the catalyst surface. Considering that AO is nonplanar and that the dimethylamino groups attached to the aromatic rings are not favorable for hydrogen bonding interactions, AO could only possibly approach the surface of $TiO_2$ NPs through the methaniminium ($=NH_2^+$) group through hydrogen bonding or electrostatic interactions.

This description is supported by the observed pH dependence. It is well known that the solution pH is effective in modifying the net charge on the surface of $TiO_2$ NPs which is known to be amphoteric [26]. The net surface charge turns from positive to negative charge at pH 6.1 [60,61]. At a solution pH lower than 6.1, the positive charge on the catalyst surface is not effective to support immobilization of AO towards the catalyst surface. On the other hand, for pH higher than 6.1, there is electrostatic attraction of the surface to AO, enhancing the photocatalysis of AO [62].

A similar observation was reported for the photocatalytic degradation of MB and RhB in the presence of $TiO_2$ NPs [27,28] or $ZrO_2$ NPs [63]. The trend of photocatalytic degradation efficiency was, therefore, that it increased from pH 7 to pH 10, as the ionic state of AO was unchanged, until the solution pH reached a pKa value (pKa 9.8–10.7) [64]. The change of ionic state of AO above pKa is also inferred by the decrease of the photocatalytic degradation efficiency at pH higher than 10.

To obtain an accurate description of the photocatalytic degradation of dyes, the reaction should be followed by liquid chromatography–mass spectroscopy or ultrafast spectroscopy [65,66], but such a detailed study of AO has not been reported. In fact, from this work, it is not clear that significant degradation products remain in the solution, as they may gasify. It is important to recall that AO has a methaniminium and two *N*-dimethylamino groups attached to its aromatic rings. Based on steady-state vibrational spectroscopy (FTIR Figure 5) and the photodegradation mechanism of related organic compounds, such as MB

and RhB, and crystal violet [65], it is proposed that oxidation of AO should form Michler's ketone, which further undergoes *N*-demethylation by successive oxidation reactions to form various intermediates, followed by destruction of the conjugated structure into small compounds, such as $CO_2$ and $NH_4^+$, as shown in Figure 7. The fact that small volatile molecular products are the final form of the photodecomposition products is supported by FTIR and UV–visible spectroscopy, because no spectroscopic evidence for larger degradation products is seen. This is a null result and yet is has significance. The assignment of the final product to gaseous small molecules is also backed up by reference [26,67].

**Figure 7.** The proposed photodegradation of Auramine O using $TiO_2$ NPs catalyst.

All of these oxidation steps, which are mediated by $OH^\bullet$ and $O_2^{-\bullet}$ radicals, would occur on or close to the immobilized dyes, where direct interactions are possible between the organic molecules and the photochemically generated radicals on the catalyst surface. A similar degradation mechanism was proposed in the electrochemical degradation of AO by Hmani et al. [67], where the final product of the oxidation was $CO_2$ gas.

## 4. Materials and Methods

### 4.1. Materials and Reagents

The chemicals used in the present experiment were analytical reagent grade of $TiO_2$ NPs and AO chloride salt ($C_{17}H_{21}N_3$.HCl; 303.83 g mol$^{-1}$; CAS: 2465-27-2) which were purchased from Sigma-Aldrich Co. (St.Louis, MO, USA) and were used without any further purification. A stock dye solution was prepared by dissolving 100 mg of powdered AO chloride in distilled water to obtain a concentration of 100 mg L$^{-1}$. Experimental solutions of a desired concentration were obtained by suitable dilutions.

### 4.2. Characterization of TiO₂ Catalyst

In this study, the commercial $TiO_2$ NPs were characterized in a previous study by Suhaimi et al. [28]. The crystalline phase of the $TiO_2$ NPs was determined based on their X-ray diffraction (XRD) pattern which was measured using an XRD-7000 (Shimadzu, Kyoto, Japan) with collimated Cu Kα radiation ($\lambda$ = 0.15418 nm). As seen in Figure S9A, the XRD pattern indicated a typical pure anatase phase with the main peak being observed at $2\theta$ = 25°. This is in good agreement with the standard XRD pattern of anatase $TiO_2$ (#JCPDS 84-1286). An SEM image scanned with an SEM-JSM-7600D (JEOL, Tokyo, Japan)

indicated that the TiO$_2$ NPs have regular spherical shapes with little agglomeration. Their size was approximately 100 nm (see Figure S9B), which is similar to the report of Amini and Ashrafi [68]. With this loose agglomeration, the TiO$_2$ NPs in a colloidal solution can be considered to have a high surface area to interact with AO molecules, thereby improving its photocatalytic activity [66].

*4.3. Photocatalysis Setup*

The experimental setup for photocatalysis was reported previously by Suhaimi et al. [28]. Briefly, AO solution (20 mL) was mixed with a few milligrams of TiO$_2$ NPs in a Petri dish with a diameter of 7.5 cm and covered with a UV-transparent glass. They were gently stirred on a temperature-controlled stage. The colloidal mixtures were irradiated from above 10 cm distance using a UV fluorescent lamp (Vilber Lourmat, 6 W, 211 mm; Marne-la-Vallée cedex 3, France). The light power was reduced using an ND filter (6.25%), so that the light power on the solution was 0.28 mW/cm$^2$. After selected irradiation times, the mixtures were centrifuged at 3000 rpm for 15 min. The filtrates were collected and analyzed by a UV–visible absorption measurement in a 1 cm cuvette cell. All absorption measurements were performed using a UV-1900 spectrophotometer (Shimadzu, Kyoto, Japan).

Prior to the photocatalysis experiments, photolysis (in the absence of TiO$_2$ NPs) and adsorption (in the dark) of AO on TiO$_2$ NPs were evaluated otherwise using the same experimental conditions. The direct photolysis of AO was monitored by irradiating the dye solution, in the absence of catalyst, with 365 nm UV light using the same irradiation geometry and power described above. After a desired irradiation time, the solution was analyzed using a UV–Vis absorption spectroscopic measurement. "Dark" adsorption of AO onto the surface of TiO$_2$ NPs was evaluated by keeping the colloidal mixture in the dark to equilibrate. After a selected contact time, the mixture was centrifuged using an Eppendorf 8504 Centrifuge (Hamburg, Germany) at 3000 rpm for 15 min. The filtrate was collected and analyzed using a UV–visible absorption measurement.

The effect of contact time was evaluated for the photocatalysis of AO in a colloidal mixture of 20 mL (9.5 ppm or equivalent to 35.06 μmol L$^{-1}$) of the dye with 5 mg TiO$_2$ NPs. The effect of the initial AO concentration was studied by adjusting the concentration to be within 9.3 μmol L$^{-1}$ and 110.8 μmol L$^{-1}$ at a constant mass of TiO$_2$ NPs (5 mg). On the other hand, the effect of the catalyst dosage was evaluated by adjusting the mass of TiO$_2$ NPs (0.5–20 mg) in the mixture with a constant initial concentration of AO (35.06 μmol L$^{-1}$). Finally, the effect of temperature was investigated from the photocatalytic degradation of AO in a mixture of 20 mL AO (35.06 μmol L$^{-1}$) and TiO$_2$ NPs (5 mg) at different temperatures from 15 °C to 40 °C.

Fourier transform infrared (FTIR) spectroscopy was used to search for large-fragment photoproducts potentially formed during the photocatalytic reaction. Here, after the photoirradiation, the colloidal mixture was centrifuged, and the precipitated solid was collected and dried in an oven at 40 °C. The vibrational spectrum of the dried solid then was recorded on an FTIR (IRPrestige-21, Shimadzu, Kyoto, Japan) in a KBr disc.

## 5. Conclusions

In this study, the photocatalytic degradation of toxic cationic Auramine O (AO) in aqueous solution on TiO$_2$ nanoparticles (NPs) under 365 nm light irradiation was investigated. Prior to the photocatalysis experiments, photolysis-alone and dark-adsorption of AO on TiO$_2$ NPs were evaluated using the same experimental conditions. From this it was found that both photolysis-alone and dark-adsorption were inefficient at reducing the aqueous burden of AO. The effects of irradiation time, initial AO concentration, and catalyst dosage on the photocatalytic degradation were evaluated in detail. The results revealed that the photodegradation kinetics of AO can be described using the Langmuir–Hinshelwood model, emphasizing that the oxidation reaction is pseudo-first order. The photodegradation rate constant is 0.048 ± 0.002 min$^{-1}$, which is slower than that of MB (0.173 ± 0.019 min$^{-1}$), due to the nonplanar structure of AO. At ambient pH, the photo-

catalytic efficiency depends on the initial concentration of AO, and a maximum efficiency as high as $96.2 \pm 0.9\%$ was achieved from a colloidal mixture of 20 mL ($17.78\ \mu mol\ L^{-1}$) AO solution in the presence of 5 mg of $TiO_2$ NPs. The photocatalytic efficiency of the dye decreases nonlinearly with increasing the initial concentration and catalyst dosage. The activation energy of the photocatalytic degradation of AO on the $TiO_2$ NPs was estimated to be $4.63\ kJ\ mol^{-1}$. The photocatalytic degradation of AO was assessed by observing the effect of pH, additional scavengers and $H_2O_2$, from which it was confirmed that the degradation is due to an oxidation reaction of the immobilized dyes on the catalyst surface, where they have direct interactions with photochemically generated $OH^{\bullet}$ and $O_2^{-\bullet}$. The nature of the degradation products of photocatalytic removal of AO was evaluated using Fourier transform infrared (FTIR) spectroscopy. The steady-state FTIR spectroscopy did not show any detectable byproducts of photocatalytic degradation of AO. This implies that, although the catalytic reaction may involve many organic intermediates, through *N*-demethylation and successive oxidation reactions, the final photocatalytic products were volatile compounds, such as $CO_2$ and $NH_4^+$, which escaped from the solution. The overall results provide detailed description of photocatalytic degradation of toxic cationic AO, a dimethylmethane fluorescent dye, in an aqueous heterogeneous solution on $TiO_2$ NPs irradiated using 365 nm light. Finally, we confirmed that using $TiO_2$ in the form of NPs greatly enhanced the rate of AO removal from solution compared to the use of micro-sized powders.

**Supplementary Materials:** The following supporting information can be downloaded online at: https://www.mdpi.com/article/10.3390/catal12090975/s1, Figure S1: UV–Vis absorption spectra following photolytic decomposition of AO as a function of irradiation time; Figure S2: Absorption spectra of AO in aqueous colloidal solution in the presence of 5 mg TiO2 NPs in the dark at different contact times; Figure S3: Images of a colloidal mixture of 20 mL of AO solution before irradiation and after irradiation; Figure S4: AO solution before irradiation and after irradiation before and after centrifugation; Figure S5: Absorption spectra of AO in aqueous colloidal mixture with 5 mg TiO2 NPs after irradiation for 30 min at different temperatures; Figure S6: Absorption spectra of different concentrations AO in an aqueous colloidal mixture with 5 mg TiO2 NPs before and after irradiation; Figure S7: Absorption spectra of different concentrations AO in aqueous colloidal mixture with different masses of $TiO_2$ NPs, and a plot of the remaining AO concentration $C_t$) as a function of the catalyst dosage; Figure S8: The plot of $\eta$ value of photocatalytic degradation of AO in aqueous colloidal mixture with 5 mg $TiO_2$ NPs at different pHs; Figure S9: XRD patterns of anatase $TiO_2$ NPs, with comparison to standard data (#JCPDS 84-1286) and SEM image of $TiO_2$ NPs at $\times$50,000 magnification.

**Author Contributions:** Conceptualization, A.U.; methodology, A.U., C.P.Y.K. and N.A.A.S.; validation, A.U., J.H., J.-W.L. and M.N.; investigation, C.P.Y.K., N.A.A.S. and N.N.M.S.; writing—original draft preparation, A.U.; writing—review and editing, A.U. and J.H.; supervision, A.U., J.H., J.-W.L. and M.N. All authors have read and agreed to the published version of the manuscript.

**Funding:** This research received no external funding.

**Acknowledgments:** Jonathan Hobley is grateful to National Cheng Kung University's NCKU90 Distinguished Visiting Scholar Program for hosting his research and to MOST for providing research funding under project number 111-2222-E-006-007.

**Conflicts of Interest:** The authors declare no conflict of interest.

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
