# Peer review of "Auramine O UV Photocatalytic Degradation on TiO2 Nanoparticles in a Heterogeneous Aqueous Solution"

_catalysts, doi:10.3390/catal12090975_

Round 1

Reviewer 1 Report

[Catalysts] Manuscript ID: catalysts-1849601

Article Title: Mechanistic Insights into Auramine O UV Photocatalytic Degradation on TiO2 Nanoparticles in a Heterogeneous Aqueous Solution

Comments and recommendations:

1.      The authors have reported the photocatalytic degradation of the dimethylmethane fluorescent dye, Auramine-O (AO), using anatase TiO2 with 100 nm particle size under 365-nm light irradiation with the final product of the oxidation, CO2 gas.

2.      I went through entire manuscript and found that the paper is logically arranged, abstract and highlights reflect the coverage of main thematic area chosen for the study. The introduction provides sufficient background and include all relevant references coverage of main thematic area chosen for the study. I am pleased to endorse the manuscript to be considered for publication in Catalysts.

3.      However, there are methods to improve TiO2 to work under visible light by shifting the TiO2 absorption to visible region. Please justify and for reference see the article published on TiO2 modification.

·         Nair, R. V., Gummaluri, V. S., Murukeshan, V. M., & Vijayan, C. (2022). A review on optical bandgap engineering in TiO2 nanostructures via doping and intrinsic vacancy modulation towards visible light applications. Journal of Physics D: Applied Physics.

·         Khan, M. S., Shah, J. A., Riaz, N., Butt, T. A., Khan, A. J., Khalifa, W., ... & Bilal, M. (2021). Synthesis and characterization of Fe-TiO2 nanomaterial: Performance evaluation for RB5 decolorization and in vitro antibacterial studies. Nanomaterials, 11(2), 436.

Author Response

Reviewer 1

  1. The abstract is not display the goal of this paper

Our Response: We have added the aim of this work in Abstract.

  1. Page 3., line 109-110, from the SEM image before and after adsorption, we cannot get the pore size of PP.

Our Response: The statement in line 109-110 might be misunderstood. The BET surface area and BJH pore size distribution were measured using a surface analyzer, not from the surface morphology recorded by SEM imaging. The surface area and pore size are too small and they cannot be precisely estimated based on the SEM images. Nevertheless, we have added the BET and BJH pore size after AB25 adsorption. 

  1. The language of the paper needs to be improved by a native English speaker!

Our Response: We have gone through the manuscript carefully, and this revised manuscript has been edited and reviewed by Jonathan Hobley, an English native speaker. For example, several examples of miss use of definite and indefinite article have been corrected. Other corrections include additional commas and punctuation; breaking up of longer sentences, etc.

Reviewer 2 Report

The manuscript title is “Mechanistic insights……”, but there was any mechanism investigation about the photocatalytic process, characterizations about catalyst, photoelectric aspect, and free radicals, which was an important deficiency for the paper. Only a few of parameter effects were investigated, and corresponding kinetics were obtained. Moreover, the TiO2 photocatalysis is not a novel topic.

Author Response

Reviewer #2

The manuscript title is “Mechanistic insights……”, but there was any mechanism investigation about the photocatalytic process, characterizations about catalyst, photoelectric aspect, and free radicals, which was an important deficiency for the paper. +Only a few of parameter effects were investigated, and corresponding kinetics were obtained.

Our response:

We agree that a single paper can never generally yield the entire mechanism for a particular photochemical system. You only have to look at the history and number of publications for systems such as spiropyran, spirooxazine, as well as the classic debate over the PICT and TICT transient state in the “Simple” photo-transformation of (dialkylamino)benzonitriles). Even now, new mechanistic insights are being provided for these, now classic, photochemical systems (studied since the 1950s). Therefore, we agree that to use the term “Mechanistic insights ….” Is probably overplaying the significance of the work. It is merely an initial study of the reaction with Auramine O on nanoparticles of TiO2. Therefore, we have altered the title of the paper. It is now “Auramine O UV Photocatalytic Degradation on TiO2 Nanoparticles in a Heterogeneous Aqueous Solution”

Moreover, the TiO2 photocatalysis is not a novel topic.

Our response:

Of course the referee is correct, but as explained above, photochemical systems are rarely elucidated in a single, or a few works. Generally, a lot of works are required to generate a consensus as to what is actually going on. We do agree that the references supplied are deficient in this respect and we add several new ones. In particular, we have added the following reference, which covers the TiO2 catalytic degradation of Auramine O on oxide powders (not nanoparticles as in our case):

[Ref No.34] https://onlinelibrary.wiley.com/doi/10.1002/(SICI)1097-4660(200003)75:3%3C205::AID-JCTB201%3E3.0.CO;2-L

There are several differences between our paper and this paper, due to the fact that the older work is not on nanoparticles. For example, in our work, we achieve the same rate of Auramine decomposition using nearly 10 times less TiO2. There are many other differences between the two works that are attributable to the use of nanoparticles. There are also, thankfully, several similarities between the works, which gives affirmative verification of many of the conclusions from the earlier work. This type of affirmation of older works, whilst pushing forward new findings is a general part of the traditional scientific approach. We are sure that most can agree that some affirmative overlap should exist between new works and older studies.

Our analysis and discussion also has many differences to the older work. In some respects the older work covers aspects that we do not and our work also covers aspects that the older work does not.

This is normal when researchers are elucidating a photochemical mechanism. It is rarely the case that a single work that is published can crack all of the mechanistic intricacies in a single shot.

We also add the following reference for photodegradation of Auramine O in ZnO, which also has interesting findings, but that is also significantly different to the current paper.

[Ref. No. 35] http://nopr.niscair.res.in/bitstream/123456789/22949/1/IJCT%208(6)%20496-499.pdf

Are all the cited references relevant to the research?

Our response:

We have added another 7 references. This deficiency was also pointed out by reviewer 1 and this was also covered in response to their comment.

Are the methods adequately described?

Our response:

Sections 4.2 and 4.3 have the same subtitle. We accept that this is confusing and we changed the section titles to “Source materials and their characterization” and “Photocatalytic studies”, respectively.

Additional experimental details were missing, and these have now been added. We thank the reviewer for pointing out these deficiencies.

Are the results clearly presented?

Our response:

We agree with the referee that the presentation was deficient. A full English and continuity edit has been performed

Are the conclusions supported by the results?

Our response:

Some corroboration of our conclusions can be found by comparing overlap between our paper and new references (34, 35). This should at least provide additional confidence in our overall conclusions, and in particular in our further conclusions that we make to advance our knowledge compared to the newly referenced works. The tenacious conclusion based on FTIR was also highlighted by referee 3 and this conclusion has been re-written.

Reviewer 3 Report

The paper with title “Mechanistic Insights into Auramine O UV Photocatalytic Degradation on TiO2 Nanoparticles in a Heterogeneous Aqueous Solution” major revision is needed before publication. Specific comments:

1)               The title should be revised, as in my opinion the work is not focused on the investigation of the mechanism but only in application.

2)               The novelty of the work should be highlighted.

3)               The specific objectives of the work should be clearly specified. Lines 102-111 should be corrected.

4)               Please correct the oxygen reactive species, according to IUPAC rules.

5)               Please correct OH‒● (Line 238 and 247).

6)               It is better to use SI units (eg. not ppm).

7)               The contribution of holes, electrons as well as hydroperoxyl radicals should also be investigated.

8)               Please explain in detail / add the data how steady state vibrational spectroscopy (FTIR Figure 5) contributed to propose the photodegradation mechanism in Fig. 7?

9)               According to authors statement in introduction section “AO and its derivatives cause long-term impacts on aquatic environments, as well as having other health risks”, thus toxicity evolution during the process should be assessed.

10)           Please add data about stability and reusability of the catalysts.

Author Response

Reviewer #3

The paper with title “Mechanistic Insights into Auramine O UV Photocatalytic Degradation on TiO2 Nanoparticles in a Heterogeneous Aqueous Solution” major revision is needed before publication. Specific comments:

  • The title should be revised, as in my opinion the work is not focused on the investigation of the mechanism but only in application.

Our response:

The title has been revised. The new title is “Auramine O UV Photocatalytic Degradation on TiO2 Nanoparticles in a Heterogeneous Aqueous Solution”

  • The novelty of the work should be highlighted.

Our response:

We have added a couple of references to previous works and we highlight differences between our work and the earlier works (34,35).

  • The specific objectives of the work should be clearly specified. Lines 102-111 should be corrected.

Our response:

We agree with the reviewer and we have changed this part to read:

“Photocatalysis of AO on semiconductor oxides has been intensively investigated, but several aspects on the photocatalytic degradation of the dye are still deficient. In particular, photochemical systems are complicated and it takes time to elucidate systems as the literature about them builds up. In general, a lot of works are required to generate a consensus as to what is actually going on.

Therefore, in this study, photocatalytic degradation of AO on anatase TiO2 NPs under 365-nm light irradiation was investigated. The objective is to systematically evaluate the effect of irradiation time, the initial concentration of AO, and catalyst dosage on the photocatalytic degradation of the dye. The efficiency and rate constant of the photodegradation were estimated based on absorption spectra of AO before and after irradiation. The photocatalytic degradation data were analyzed with standard empirical models. The thermodynamics of the AO photodegradation process were assessed by monitoring the effect of temperature. The photocatalytic degradation mechanism was further assessed by observing the effect of pH and additional scavengers as well as H2O2.

This work should provide a baseline for future works, which may include using doped and sensitized TiO2 in order to shift the absorbance further to the visible to improve catalytic efficiency [36-40].  It is important to highlight that there are several differences between this study and those in literature, including the use of NPs, which should give better photocatalytic degradation rates. There are also several similarities and some agreements between this current study and the reported works, which gives affirmative verification of many of the conclusions from the earlier work, which is a general duty of the traditional scientific approach.”

  • Please correct the oxygen reactive species, according to IUPAC rules.

Our response:

We have corrected the notation of hydroxyl (OH) and oxygen () radicals according to IUPAC rules throughout the revised manuscript.

  • Please correct OH‒●(Line 238 and 247).

Our response:

We have corrected the mistakes.

  • It is better to use SI units (eg. not ppm).

Our response:

We have deleted the “ppm” unit.

  • The contribution of holes, electrons as well as hydroperoxyl radicals should also be investigated.

Our response:

We thank reviewer for her/his comment and suggestion. As has been well documented in literature, solvated holes and electrons in aqueous solution have very short lifetimes (a few hundred fs) which are much faster than diffusional dynamic of dyes in the solution, it is therefore reasonable to consider that that holes and electrons in aqueous solution form  and OH radicals and that the degradation of dyes were mediated by these radicals.

On the other hand, hydroperoxyl () radical is formed through the formation of  radical. Thus, although the role of  radical in the degradation of AO is not negligible, it could be represented by considering  radical.

  • Please explain in detail / add the data how steady state vibrational spectroscopy (FTIR Figure 5)

contributed to propose the photodegradation mechanism in Fig. 7?

Our response:

We have highlighted why the null result for spectroscopic evidence of photoproducts is significant in deriving the mechanism;

”The fact that small volatile molecular products are the final form of the photodecomposition products is supported by FTIR and UV-Visible spectroscopy, because no spectroscopic evidence for larger degradation products is seen. This is a null result and yet is has significance. The assignment of the final product to gaseous small molecules is also backed up by reference [34,69].”

  • According to authors statement in introduction section “AO and its derivatives cause long-term

impacts on aquatic environments, as well as having other health risks”, thus toxicity evolution during

the process should be assessed.

Our response:

Unfortunately, this experiment is rather involved and would require a different experimental protocol. We hope that the referee can accept that this very important suggestion should become a paper in itself for a toxicology paper.

10) Please add data about stability and reusability of the catalysts.

Our response:

Unfortunately, we did not carry out any regeneration experiments of the TiO2 NPs as photocatalyst, but we have collected and characterized the photocatalyst after the photocatalytic degradation of AO (see Fig. S3). We found that FTIR spectrum and XRD pattern of the retrieved TiO2 NPs are similar to those of the photocatalyst before photocatalysis, confirming the stability of the photocatalyst

Round 2

Reviewer 2 Report

It could be accepted.